# X-ray radiation damage cycle of solvated inorganic ions

Dana Bloß [1] ✉, Florian Trinter [2,3], Isaak Unger[4], Christina Zindel [1], Carolin Honisch[1], Johannes Viehmann[1], Nils Kiefer[1], Lutz Marder [1], Catmarna Küstner-Wetekam [1], Emilia Heikura[1], Lorenz S. Cederbaum [5], Olle Björneholm[4], Uwe Hergenhahn [2], Arno Ehresmann [1] & Andreas Hans [1] ✉

X-ray-induced damage is one of the key topics in radiation chemistry. Substantial damage is attributed to low-energy electrons and radicals emerging from direct inner-shell photoionization or produced by subsequent processes. We apply multi-electron coincidence spectroscopy to X-ray-irradiated aqueous solutions of inorganic ions to investigate the production of low-energy electrons (LEEs) in a predicted cascade of intermolecular charge- and energy-transfer processes, namely electron-transfer-mediated decay (ETMD) and interatomic/intermolecular Coulombic decay (ICD). An advanced coincidence technique allows us to identify several LEE-producing steps during the decay of 1s vacancies in solvated $Mg^{2+}$ ions, which escaped observation in previous non-coincident experiments. We provide strong evidence for the predicted recovering of the ion's initial state. In natural environments the recovering of the ion's initial state is expected to cause inorganic ions to be radiation-damage hot spots, repeatedly producing destructive particles under continuous irradiation.

The interaction of ionizing radiation like soft X-rays with aqueous solutions is a key for understanding the radiation damage to biological systems on a molecular level. While the macroscopic consequences of radiation exposure are rather well understood and risks may be quantified depending on the received dose (e.g., enhanced cancer risk)[1], detailed knowledge about the cascade of mechanisms happening after a single photon-matter interaction in complex environments is still limited[2–5].

Besides the direct damage caused by the absorption of photons, i.e., inner-shell photoionization, an even more important role is assigned to indirect damage through highly reactive photoproducts and low-energy electrons (LEEs) resulting from secondary processes[3,4]. Such LEEs with energies typically below 30 eV are known to be genotoxic, e.g., by causing irreparable double or multiple strand breaks of

the DNA[3,6]. One source of LEEs are inelastic-scattering events of the primary photoelectrons and fast Auger electrons[3].

In a recent pioneering work, Stumpf et al. predicted that LEEs may also be emitted very efficiently and locally at the site of ionization of inorganic ions by the intermolecular charge- and energy-transfer processes electron-transfer-mediated decay (ETMD) and interatomic/ intermolecular Coulombic decay (ICD)[7]. Both processes are known to emit LEEs from, e.g., water dimers[8], larger water clusters[9], liquid water[10–12], aqueous solutions[13], or solvated dielectrons[14].

In their work, Stumpf et al. chose the $Mg^{2+}$ ion as example because of its importance in biochemistry[7]. Mg is one of the most relevant elements in the biosphere and plays a key role in many important processes and functions, such as nerve conduction, energy generation of the cells, cell membrane regulation, or DNA stabilization[15,16]. Further,

[1]Institute of Physics and Center for Interdisciplinary Nanostructure Science and Technology (CINSaT), University of Kassel, Kassel, Germany. [2]Fritz-Haber-Institut der Max-Planck-Gesellschaft, Berlin, Germany. [3]Institut für Kernphysik, Goethe-Universität Frankfurt, Frankfurt am Main, Germany. [4]Chemical and Biomolecular Physics, Department of Physics and Astronomy, Uppsala University, Uppsala, Sweden. [5]Theoretical Chemistry, Institute of Physical Chemistry, University of Heidelberg, Heidelberg, Germany. ✉e-mail: dana.bloss@uni-kassel.de; hans@physik.uni-kassel.de

as a metal, Mg is particularly sensitive to X-ray irradiation due to its large photoionization cross section compared to the biologically more abundant elements H, C, N, and O.

We investigated X-ray-irradiated solvated $Mg^{2+}$ ions using a liquid microjet with the goal to identify the predicted ETMD and ICD processes.

After X-ray ionization of the 1s level of a solvated $Mg^{2+}$ ion, Auger decay creates a $Mg^{4+}$ ion with two vacancies in the $n = 2$ shell. The subsequent ETMD and ICD processes involving the aqueous environment are illustrated in Fig. 1. For the most abundantly populated $Mg^{4+}$ ($2s^2 2p^4$) configurations, ETMD is the only open decay channel. Two variants are possible, namely ETMD(2) or ETMD(3)[17]. In Fig. 1, ETMD(3) is sketched exemplarily. One vacancy in the $Mg^{4+}$ valence shell is filled by an outer-valence electron from a water molecule and a second outer-valence electron, named ETMD electron, from yet another water molecule is ejected. For ETMD(2), the electron filling the vacancy and the electron being emitted originate from the same water molecule. Excited ionic Auger final states, such as $Mg^{4+}$ ($2s^1 2p^5$) or $Mg^{4+}$ ($2s^0 2p^6$) may decay by ICD. In ICD, a 2p electron fills a 2s hole and the released energy ionizes an outer-valence electron from a neighboring water molecule, termed ICD electron.

ETMD is a charge-transfer process and reduces the charge of the $Mg^{4+}$ ion to $Mg^{3+}$. In contrast, ICD is an energy-transfer process and can be described as a virtual-photon exchange via dipole transitions[18]. ICD is the dominant process, if both ICD and ETMD are energetically

possible. This is reflected in the lifetimes of these processes, which e.g., for small van der Waals clusters are typically on the femtosecond time scale for ICD vs. picosecond time scale for ETMD[19]. Interestingly, the ICD and ETMD processes at play for solvated Mg ions in water were calculated to exhibit lifetimes below 1 fs for ICD and below 20 fs for ETMD, both being surprisingly fast[7]. This is attributed to the nature of hydrogen bonds and the presence of several water molecules increasing the decay probabilities.

In their theoretical investigation, Stumpf et al. calculated the decay cascade in a $Mg^{2+}$-$(H_2O)_6$ cluster, which served as a model for aqueous solutions[7]. A part of this cascade is displayed in Fig. 2 starting with the inner-shell-photoionized $Mg^{3+}$ ($1s^{-1}$) ion. For clarity the figure considers only major decay routes, accounting for 93% of all pathways. The first steps of the cascade are various Auger decays indicated with solid black arrows, leading to $Mg^{4+}$ ($2s^{-1}2p^{-1}$ [$^3P$]), $Mg^{4+}$ ($2s^{-1}2p^{-1}$ [$^1P$]), or $Mg^{4+}$ ($2p^{-2}$ [$^1D$, $^1S$]) states. Only a minor fraction (below 4%) of the inner-shell holes decays directly non-locally by core-level ICD and ejection of an electron from neighboring water[7,20]. A similar small fraction is expected to decay via fluorescence[5,21].

For the $Mg^{4+}$ ($2s^{-2}$ [$^1S$]) (very weak, not shown in Fig. 2) and $Mg^{4+}$ ($2s^{-1}2p^{-1}$ [$^1P$]) Auger final states, the ICD channel is open. All other states will decay further by ETMD[7]. The ETMD final states are either $Mg^{3+}$ ($2s^{-1}$ [$^2S$]) or $Mg^{3+}$ ($2p^{-1}$ [$^2P$]). The $Mg^{3+}$ ($2s^{-1}$ [$^2S$]) state undergoes ICD ending in the $Mg^{3+}$ ($2p^{-1}$ [$^2P$]) state as well.

Finally, also the $Mg^{3+}$ ($2p^{-1}$ [$^2P$]) state decays by ETMD[7], recovering the initial charge state of the Mg ion before the photoionization. Remarkably, the lifetimes of all steps are predicted to be extremely short and even for the ETMD steps in the femtosecond range, resulting in an overall lifetime of 220 fs for the complete cycle[7]. Assuming continuous exposure, the same cascade could start over and over again on a very short time scale. In summary, one inner-shell ionization is predicted to result on average in the emission of one fast Auger electron, 2.4 LEEs, and 4.3 water radicals[7].

While ETMD after valence ionization of solvated inorganic ions has been investigated before[13,22], a study concerning the decay of Auger final states has been reported only recently[5]. Gopakumar et al. used a hemispherical electron analyzer to explore the decay of 1s-ionized $Al^{3+}$ in aqueous solution[5]. Two LEE features were observed and attributed to ETMD[5]. For Mg, however, no difference compared to the pure water reference was observed. A general challenge in electron spectroscopy on liquids is the large intensity in the low-kinetic-energy part of the spectrum, caused by inelastically scattered electrons[23–25]. Elimination of this background is practically impossible for hemispherical electron spectrometers. It remained unclear from Ref. 5, whether an X-ray-induced cascade of ICD and ETMD can be observed for $Mg^{2+}$ at all and if not, whether there is a physical explanation for it or whether it is just masked by the low-energy background.

In this work, we investigated the photoemission of Mg ions in aqueous solution by using multi-electron coincidence detection. This technique provides two valuable advantages vs. non-coincident electron spectroscopy with a hemispherical analyzer. Firstly, it enables a significant reduction of the LEE background[23]. Secondly, different initial states of the cascade can be probed by setting a coincidence condition to the respective photoelectron: the full cascade after 1s ionization (magenta background in Fig. 2), direct ionization of the 2p (green background), or the 2s state (blue background). This allows us to disentangle specific processes by considerably reducing the complexity of the observed decay route.

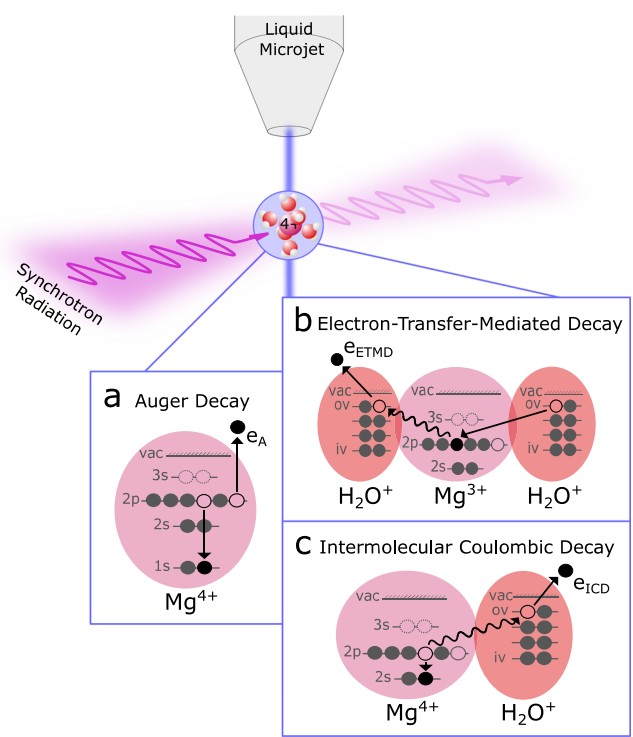

Fig. 1 | **Local and non-local processes in $Mg^{2+}$ ions.** Sketch of the local Auger decay as well as the intermolecular electron-transfer-mediated decay (ETMD) and interatomic/intermolecular Coulombic decay (ICD) in Mg ions (magenta spheres) surrounded by water molecules (red spheres). Different involved energy levels of the ion (1s, 2s, 2p, and 3s) and of the molecule [inner-valence (iv) and outer-valence (ov)] are depicted including the vacuum level (vac). Solvated magnesium occurs as $Mg^{2+}$ ions. $Mg^{2+}$ 1s photoionization results in $Mg^{3+}$ ions with a vacancy in the $n = 1$ shell. The inset **a** visualizes the local Auger decay of the core-ionized Mg ion emitting an Auger electron ($e_A$). Inset **b** shows a subsequent ETMD process ionizing two neighboring water molecules, reducing the charge of the metal to $Mg^{3+}$ and emitting an electron ($e_{ETMD}$). Additionally, excited final states of the Auger decay can also decay by ICD (inset **c**), producing a single water vacancy and a free electron ($e_{ICD}$).

## Results

A major challenge for the interpretation of the low-energy part of a typical electron spectrum measured from the liquid phase is the monotonously increasing, structureless signal towards low kinetic energies[23]. Without further distinction, this signal contains all processes producing LEEs. It includes, therefore, (1) photo- and Auger

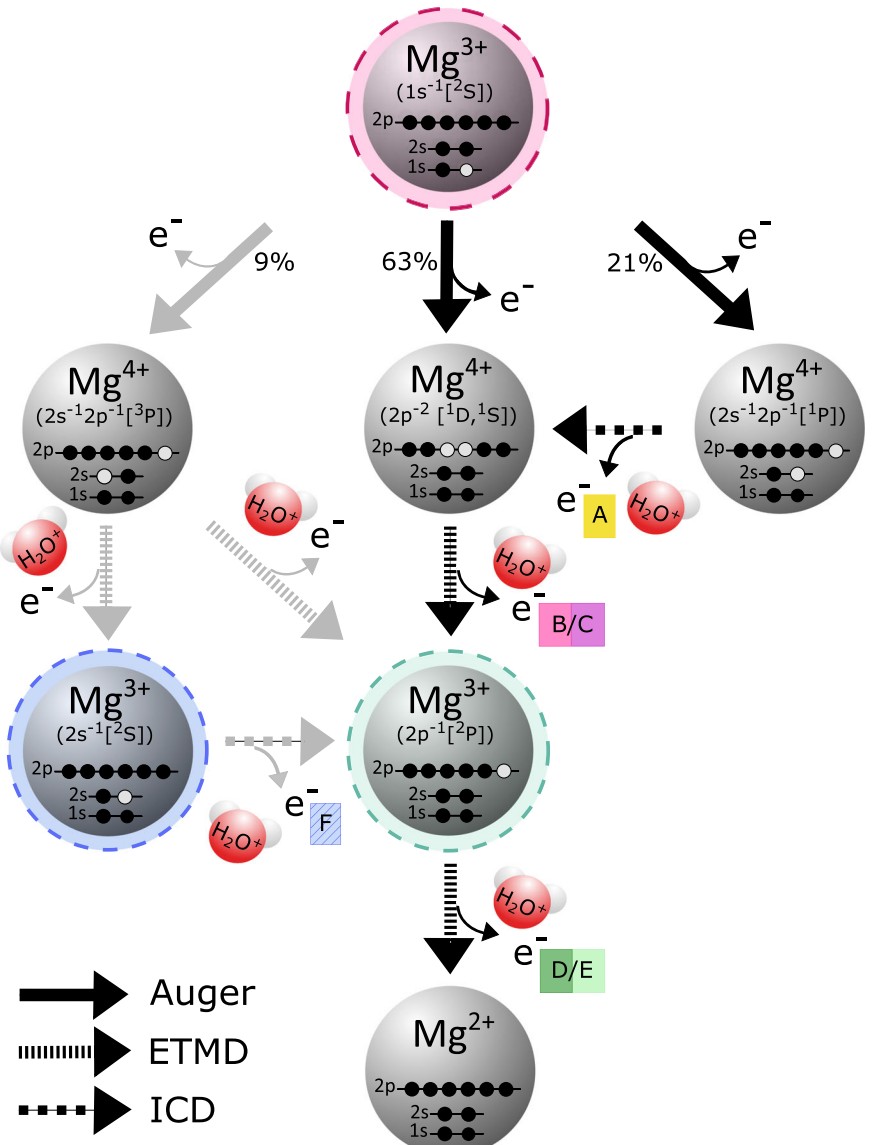

**Fig. 2 | Predicted decay cascade of Mg²⁺ ions after inner-shell ionization.** Main pathways of the predicted decay cascade of an inner-shell-ionized Mg³⁺ (1s⁻¹) ion (uppermost gray sphere with magenta background) embedded in a (H₂O)₆ cluster (modified from Ref. 7). The cascade starts with Auger decays into different possible final states. The Auger final states decay further by different electron-transfer-mediated decay (ETMD) and interatomic/intermolecular Coulombic decay (ICD) processes, ionizing the neighboring water molecules (red spheres). Electrons emitted by these processes are color coded and labeled with letters A to F, where A and F result from ICD, and B, C, D, and E from ETMD. At the end of these cascades, the Mg ion ends up in its doubly charged initial state. The Auger lifetimes are calculated to be about ~2 fs, ICD lifetimes below 1 fs, and ETMD lifetimes below 20 fs[7]. Decay steps D-F, taking place in already partly neutralized Mg centers, can also be initiated via direct 2s or 2p photoionization of the Mg ion into the Mg³⁺ (2s⁻¹) (blue) or Mg³⁺ (2p⁻¹) state (green), respectively, without involving an Auger decay. The gray arrows indicate a minor decay pathway, while the black arrows correspond to the main decay pathways after 1s ionization. Other minor Auger channels, accounting for 7% of all pathways, as well as the (unoccupied) Mg 3s level were omitted for clarity.

electrons, being inelastically scattered by the dense medium and having lost a significant amount of their kinetic energy[12], (2) secondary electrons produced by electron-impact ionization[11,12,23,24], and (3) electrons created by ETMD or ICD[12,13]. One promising approach to disentangle this signal is the detection of electron pairs or triples in coincidence[22]. Here, defined pathways may be determined, if at least one electron with a distinct kinetic energy can be identified, e.g., the photoelectron.

Figure 3a displays the LEE spectra from double-electron coincidences obtained for a 3 M MgCl₂ aqueous solution at 145 eV photon energy. The gray dashed line shows the LEE spectrum as measured without a coincidence condition, dominated by the structureless signal towards low kinetic energies. For the green solid line, we have screened the signal by accounting only for LEEs detected in coincidence with the

2p photoelectron, the resulting trace was re-normalized for better visibility. Although first distinct structures appear in the spectrum, the congested signal still cannot fully be disentangled.

We, therefore, used different subtraction methods to further separate the unstructured background from spectral features of interest. All difference spectra are achieved from subtracting a background spectrum dominated by the unstructured signal (e.g., gray dashed line) from the LEE spectrum obtained from applying a coincidence condition (e.g., green solid line). A detailed description of the normalization and subtraction procedure is provided in the section Supplementary Note 1 in the Supplementary Information. It is evident that our approach does not yield absolute or relative intensities, but all features discussed here with respect to their energetic positions in the spectra can be reliably and reproducibly deduced from the raw data

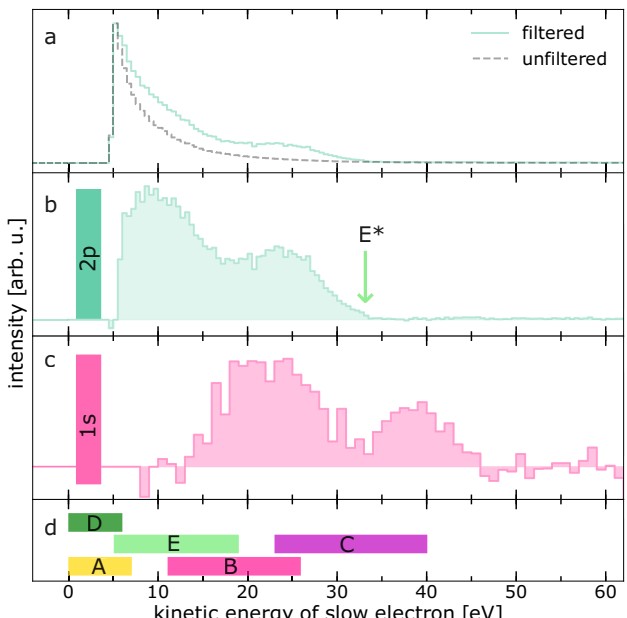

**Fig. 3 | Low-kinetic-energy electron (LEE) spectra of a 3 M MgCl₂ solution after Mg 2p and Mg 1s ionization.** a Gray dashed line: unfiltered LEE spectrum from double-electron coincidences at an exciting-photon energy of 145 eV. Green solid line: spectrum of the same dataset screened for LEEs in coincidence with the 2p photoelectron (around 89 eV kinetic energy). Both spectra are normalized to their maximum. **b** Difference spectrum of the two spectra in panel **a**. The green arrow labeled E* marks the highest estimated energy for the electron-transfer-mediated decay (ETMD) electrons e_E, including no Coulomb repulsion. **c** Difference spectrum obtained from double-electron coincidences at an excitation energy of 1387 eV and using detection of the 1s photoelectron as a coincidence filter. **d** The colored bars represent the predicted ETMD and interatomic/intermolecular Coulombic decay electron energy ranges predicted in Ref. 7 (labeled A to E), assigned to the transitions in Fig. 2 and listed in Table 1. Signal intensities in panels **b** and **c** were scaled to improve visibility. All experimental spectra for technical reasons have a low-energy cutoff of a few eV (see text). Source data are provided as a Source Data file.

### Table 1 | Predicted LEE emission after Mg²⁺ 1s ionization

| Process | Decay | Transition | Electron energy range |
|---|---|---|---|
| A | ICD | $Mg^{4+} (2s^{-1}2p^{-1}) \rightarrow Mg^{4+} (2p^{-2})$ | 0-7 eV[7] |
| B | ETMD(2) | $Mg^{4+} (2p^{-2}) \rightarrow Mg^{3+} (2p^{-1})$ | 11-26 eV[7] |
| C | ETMD(3) | $Mg^{4+} (2p^{-2}) \rightarrow Mg^{3+} (2p^{-1})$ | 23-40 eV[7] |
| D | ETMD(2) | $Mg^{3+} (2p^{-1}) \rightarrow Mg^{2+}$ | 0-6 eV[7] |
| E | ETMD(3) | $Mg^{3+} (2p^{-1}) \rightarrow Mg^{2+}$ | 5-19 eV[7] |
| F | ICD | $Mg^{3+} (2s^{-1}) \rightarrow Mg^{3+} (2p^{-1})$ | 15-24 eV |

Predicted LEE emissions (A to F) resulting from either ICD or ETMD processes after Mg²⁺ 1s photoionization listed with their respective initial and final states. The predicted energy ranges for A to E are taken from Ref. 7. For details on the estimate for process F see text.

independent of the experimental conditions (see Supplementary Figs. 2, 3, 4, and 5 in the Supplementary Information).

As a validation of our method, we firstly investigated Al³⁺ after 1s inner-shell ionization. We satisfactorily reproduced the features around 48 and 66 eV reported from recent experiments using a hemispherical electron analyzer[5] (see Supplementary Fig. 1 in the Supplementary Information).

### Ionization of Mg²⁺ 2p and 1s electrons

In Fig. 3b and c, we present the LEE difference spectra after ionization of Mg²⁺ 2p and 1s electrons with exciting-photon energies of 145 and 1387 eV and binding energies of 55.8/55.5 eV[26] and 1309.9 eV[20], respectively. These difference spectra result from double-electron

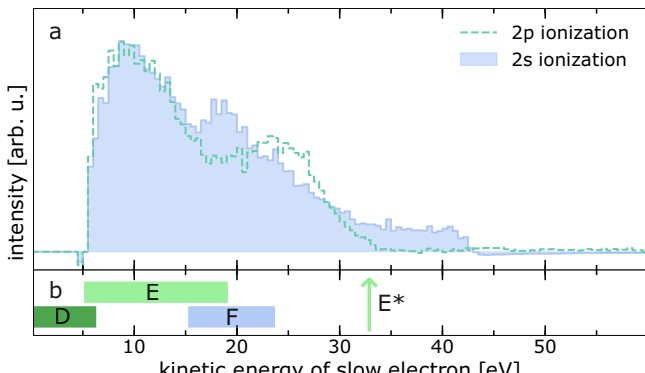

**Fig. 4 | Low-kinetic-energy electron spectra of a 3 M MgCl₂ solution after Mg 2p and Mg 2s ionization.** a Difference spectrum from double-electron coincidences at an excitation energy of 145 eV and a coincidence condition for the 2s photoelectron (blue shaded curve). For comparison the difference spectrum obtained after the 2p photoelectron coincidence filter from Fig. 3b is shown again, now as green dashed line. Both curves are normalized to their maximum. **b** The green bars (labeled D and E) are indicating the calculated electron-transfer-mediated decay (ETMD) electron energy ranges[7] for the decay of the Mg³⁺ (2p⁻¹ [²P]) state. The green arrow labeled E* marks the highest estimated energy for the ETMD electrons e_E including no Coulomb repulsion. The blue bar (labeled F) represents an estimate for the energy of the interatomic/intermolecular Coulombic decay electron e_F (see text). Source data are provided as a Source Data file.

coincidences applying the coincidence condition with the respective photoelectron (2p or 1s) and using normalization and background subtraction (see section Supplementary Note 1 in the Supplementary Information). Line colors correspond to the background color of different steps in the various decay steps in the theoretical reaction scheme (Fig. 2). For comparison, in Fig. 3d, the predicted energy ranges of ETMD and ICD electrons emitted in different steps (labeled A to E, see Table 1) in the cascades after the 1s ionization are indicated as bars[7]. A green arrow labeled E* in Fig. 3b represents an estimate of the highest possible ETMD(3) electron energy (~ 33.2 eV) resulting from the ionization energies in the initial and final states and fully neglecting any Coulomb repulsion.

The green shaded curve of Fig. 3b exhibits two features at around 10 and 24 eV and a high-energy cutoff at about 33 eV. The magenta shaded curve of Fig. 3c shows two features centered around 22 and 40 eV, the latter of which extends up to 46 eV. The low-energy cutoffs of the curves result from the application of a retardation bias voltage at the magnetic bottle (see section Supplementary Note 3 in the Supplementary Information).

For all cases, as a consistency check, datasets were also recorded at the slightly different exciting-photon energies of 175, 1367, and 1397 eV. The features in the spectra (shown in Supplementary Figs. 2, 3, 4, and 5 in the Supplementary Information) do not show any significant energetic shifts dependent on the photon energy, indicating that they indeed originate from photon-energy-independent ETMD- or ICD-like processes.

### Ionization of Mg²⁺ 2s electrons

Besides the above-mentioned 2p and 1s ionization of Mg we received the spectrum after Mg 2s ionization as well. The blue shaded curve in Fig. 4a shows the difference spectrum of double-electron coincidences after 2s ionization at an exciting-photon energy of 145 eV. The coincidence condition is set for the 2s photoelectron and the background has been subtracted as discussed above. The green dashed line shows the corresponding difference spectrum after 2p electron ionization copied from Fig. 3b. The blue shaded curve exhibits signal between 6 eV and approximately 35 eV with a maximum around 19 eV. Above 35 eV, no significant feature can be distinguished. As evident from

Fig. 2, after 2s ionization, additionally to ETMD from $Mg^{3+}$ ($2p^{-1}$ [$^2P$]) states, every ionization event triggers emission of an ICD electron originating from the $Mg^{3+}$ ($2s^{-1}$ [$^2S$]) → $Mg^{3+}$ ($2p^{-1}$ [$^2P$]) transition (process F of Fig. 2), which is only a minor channel after 1s ionization. ICD after direct 2s ionization has already been reported in Ref. 26.

Figure 4b illustrates the predicted LEE energies expected after 2s and 2p electron ionization. The blue box is an estimate of the excess energy of the ICD electron from the $Mg^{3+}$ ($2s^{-1}$ [$^2S$]) → $Mg^{3+}$ ($2p^{-1}$ [$^2P$]) transition including the binding energies of the $Mg^{2+}$ 2s (94.3 eV[26]) and 2p states (55.8 and 55.5 eV[26]) in solution, the water valence states (around 11.3 to 17.3 eV[27,28]), and a potential Coulomb energy between the resulting ions of 4 to 6 eV[20], which was theoretically determined for $Mg^{2+}$ in solution. The expected ICD electron energy range is between 15 and 24 eV.

The green bars for processes D and E again indicate the predicted energy range for the ETMD electrons from the $Mg^{3+}$ ($2p^{-1}$ [$^2P$]) decay[7]. The green arrow labeled E* again shows the estimated highest possible ETMD(3) electron energy.

## Discussion

We start with a comparison of the LEE difference spectrum in Fig. 3b and the corresponding predicted ETMD(2) and ETMD(3) electron energy ranges for the ETMD processes D and E (Fig. 3d). The difference spectrum corresponds to the decay of 2p-ionized $Mg^{3+}$. An assignment of the two experimentally observed features to ETMD(2) and ETMD(3) seems evident, although the energy discrepancy between experiment and prediction is significant. While the calculations were performed on a $Mg^{2+}$-$(H_2O)_6$ cluster and therefore considered only the first solvation shell, the full solvation in the experiment may lead to higher observable kinetic energies. This is due to larger polarization screening or charge separation over a wider range in the liquid phase beyond the first solvation shell[29]. It has been observed earlier that in the liquid phase the screening of charge can be quite efficient[29]. Another important aspect is that in the calculations of Ref. 7 the environment of the Mg has been assumed to be neutral for each step. In a real cascade, however, the already produced $H_2O^+$ cations may influence the further steps to a certain extent, until they will be replaced by neutrals from the environment. The presence of full solvation may also influence the decay widths of the excited states, although energy and charge exchange with the first solvation shell have been found to dominate for other systems[30].

Surprisingly, there is a good agreement between the estimated high-energy cutoff of ~33.2 eV neglecting the Coulomb repulsion for the ETMD(3) electron $e_E$ (indicated by a green arrow and labeled E* in Fig. 3b) and the high-energy cutoff in the experimental spectra around at least 30 eV in the double-electron coincidences. A similar observation was reported for the spectra of the decay of inner-shell-ionized $Al^{3+}$, see Ref. 5. This is a strong evidence that not only the first solvation shell participates in the decay and therefore the charge may be delocalized or screened effectively by the extended environment[31,32].

For the $Mg^{2+}$ 1s photoionization (Fig. 3c) the presence of all steps of the cascade displayed in Fig. 2 is expected. The main contributions are the ICD electron $e_A$ and the ETMD electrons $e_B$ to $e_E$. There is strong plausibility to assign the peak at the highest kinetic energies (around 40 eV) in the difference spectrum in Fig. 3c to the ETMD(3) electrons $e_C$ emitted in the $Mg^{4+}$ ($2p^{-2}$ [$^1D$, $^1S$]) decay. Their predicted ($e_C$ in Fig. 3d) and measured energies match relatively well, and no other decay step is expected to emit more energetic electrons. This interpretation is supported by the fact that this feature in the region around 40 eV appears only after 1s ionization, which makes the $Mg^{4+}$ ($2p^{-2}$ [$^1D$, $^1S$]) ETMD initial state accessible.

The second maximum in the difference spectrum of Fig. 3c, at lower kinetic energies and peaking at about 22 eV, is much more difficult to assign. It is expected to contain contributions from ETMD(2) of the $Mg^{4+}$ ($2p^{-2}$ [$^1D$, $^1S$]) states as well as a superposition of the electrons considered in the scenarios above ($e_A$, $e_B$, $e_D$, and $e_E$).

By comparing the difference spectra after 1s and 2p ionization we can investigate the impact of a neutral vs. an ionized water environment of the $Mg^{2+}$ ion on the emitted ETMD electron energies. Both pathways populate the $Mg^{3+}$ ($2p^{-1}$ [$^2P$]) state which decays via ETMD(2) or ETMD(3) emitting $e_D$ or $e_E$. In the 1s ionization case, on average 1.4 ionized water molecules are created prior to the ETMD, while in the 2p case only neutral water is around. For the present data, however, it seems that the superposition of several contributions in the peak around 22 eV of the 1s spectrum prevents any reliable conclusion about this effect.

Nevertheless, to obtain a better understanding of the feature around 22 eV, we experimentally initiated the cascade displayed in Fig. 2 in yet an alternative way, namely by ionizing a $Mg^{2+}$ 2s electron. This pathway occurs with only minor probability after the relaxation of a $Mg^{3+}$ ($1s^{-1}$) inner-shell vacancy via Auger decay. Now, an additional ICD channel (see Fig. 2) as well as ETMD(2) or ETMD(3) electrons ($e_D$ and $e_F$) are expected. The agreement of the estimated energies of the ICD electron ($e_F$) corresponding to the $Mg^{3+}$ ($2s^{-1}$ [$^2S$]) → $Mg^{3+}$ ($2p^{-1}$ [$^2P$]) transition shown as a blue bar (labeled F) in Fig. 4b and the maximum in the blue curve of Fig. 4a around 19 eV is remarkable. This maximum only appears after 2s ionization and is absent in the 2p ionization difference spectrum (green dashed line in Fig. 4a). It seems straightforward to assign this feature to the emitted ICD electrons.

For the ETMD electrons resulting from the second-step decay we indeed find signal in the blue shaded curve from 5 eV extending above the background level to about 35 eV, comparable to the pure ETMD difference spectrum (green dashed line, copied from Fig. 3b). The two ETMD features, clearly visible in the case of the 2p ionization, may be strongly disturbed in the case of the 2s ionization by the preceding ICD. The latter produces an additional $H_2O^+$ ion close by and may thus introduce a stronger Coulomb repulsion to the system, therefore shifting the ETMD electrons to lower kinetic energies. The feature around 10 eV is mainly the analogue to the feature at equal kinetic energies in Fig. 3b. It is expected to appear due to ETMD(2) from 2p-ionized states which are populated by ICD of the 2s-ionized states. Its low-energy onset cannot reliably be deduced from the present data due to the applied retardation voltage.

The predicted energies of the ETMD electrons[7] shown as green bars (labeled D and E) in Fig. 4b exhibit a significant discrepancy, as was discussed above in the solely 2p ionization case. In all steps of the cascade, besides ionization of neighboring water, the Cl counter ion could in principle participate in the decay. However, in earlier studies it was found that for $Mg^{2+}$ in solution up to a concentration near the saturation for $MgCl_2$ no contact ion pairing can be found in the first solvation shell[20,33,34]. Hence, we expect the contribution from Cl to the ICD and ETMD processes to be negligible.

We presented multi-electron coincidence spectra after 2p, 2s, and 1s electron photoionization of $Mg^{2+}$ from a 3 M aqueous $MgCl_2$ solution and compared the results to the predicted decay cascade after $Mg^{2+}$ 1s electron inner-shell ionization[7]. Here, the absorption of a single high-energy photon by the metal ion leads to an ultrafast, radiationless decay cascade producing several LEEs via ETMD and ICD that potentially may cause local damage to the surrounding of the ion. Even more important, the metal ion ends up in its initial state rapidly, ready for the absorption of another photon. Consequently, the cascade can start over and over again, multiplying the local damage massively. The produced ionized water molecules are expected to be transferred to a further solvation shell due to Coulomb explosion, with new water molecules diffusing into the surrounding of the metal ion, keeping the charge- and energy-transfer channels open. This decay cascade is not exclusive to Mg ions, but could proceed after ionization of other solvated metal ions as well.

Interpretation of the experimental LEE spectra is challenging because of the high background of slow electrons. In our study the application of coincidence conditions, subsequent normalization, and background-subtraction procedures, however, reveals significant structures in the LEE spectra. The coincidence technique enabled

assignments of individual spectral features to certain steps in the decay pathways, an information that is inaccessible by other experimental techniques due to the congested spectrum. We envision our results to stimulate further efforts for the development of spectroscopic methods on liquids as well as for the refinement of theoretical models to improve the agreement between theory and experiment.

## Methods

A cooled (4 °C) liquid microjet[35–37] with a 30 μm glass nozzle and a constant flow rate between 0.6 and 0.8 ml/min was used for target delivery. Ground potential or a low bias voltage could be applied to the sample via a gold wire, reducing the streaming potential. In vacuum, the microjet was crossed orthogonally with synchrotron radiation and collected at a cold trap filled with liquid nitrogen.

The presented data were obtained during two beamtimes at synchrotron radiation sources. The synchrotron radiation was provided by the U49-2_PGM−1 beamline[38] at BESSY II in Berlin operating in single-bunch mode or the P04 beamline[39] at PETRA III in Hamburg operating in 40-bunch mode. The first beamline provides a focus size of around 25 μm (vertical) × 85 μm (horizontal) and a temporal spacing between the light pulses of 800 ns[38]. The latter provides a focus size of approximately 20 μm (vertical) and 20 μm (horizontal) and a time spacing of 192 ns[39].

Practically, detection of two or more electrons from a single ionization event in coincidence requires a multiplexed acquisition with respect to both solid angle and electron kinetic energies. The kinetic energy of the emitted electrons was therefore measured with a magnetic bottle time-of-flight spectrometer[22,40], which has a large acceptance angle and is mounted vertically, i.e., orthogonally to the liquid jet and the light axis. Opposite to the drift tube of the magnetic bottle, a samarium-cobalt (SmCo) permanent magnet with an additional truncated iron cone, mounted on a x-y-z manipulator, guided the emitted electrons towards the drift tube. The drift tube itself has an opening aperture of about 6 mm. Two solenoids guide the electrons via a weak but homogeneous magnetic field to the end of the drift tube, terminated by a copper mesh. The electrons can be accelerated or retarded by a voltage applied to the drift tube, and are detected by a Chevron stack microchannel plate (MCP) detector (Hamamatsu) mounted behind the copper mesh. An MCP arrangement suitable for high-pressure conditions (up to $10^{-2}$ mbar) was chosen (Hamamatsu F14844 data sheet). Similar magnetic bottle time-of-flight spectrometers have a resolution of E/ΔE around 30[40]. The temporal resolution of the experiment is determined by the duration of the synchrotron pulses, which is around 100 ps.

Electron pulses from the detector were amplified (FTA 810, EG&G), processed in a constant fraction discriminator (CFD8c, RoentDek), and acquired by a time-to-digital converter (TDC8HP, RoentDek). The TDC was triggered with a reference clock synchronized to the bunch pattern of the storage ring. To ensure operation in a regime with negligible random coincidences, the count rate was kept low (around 1.1 kHz for double-electron coincidences and around 13 kHz for single-electron rate) compared to the repetition rate of the synchrotrons (about 1 to 5 MHz).

For preparation of the aqueous solutions, $MgCl_2$ (Alfa Aesar, 99%) and $AlCl_3$ (Alfa Aesar, 99%) were dissolved in water. For reference measurements 50 mM NaCl was added to a sample of pure water to maintain electrical conductivity. The solutions were degassed and filtered before use. The conversion of flight times to kinetic energies was done via reference measurements of the O 1s photoelectron of water. Typical acquisition times of the presented spectra were between 60 and 360 minutes.

## Data availability

The data generated in this study have been deposited in a Zenodo database [https://doi.org/10.5281/zenodo.10910949][41]. Source data are provided with this paper.

## Code availability

For the data evaluation freely available, common python packages were used. The developed code is available from the corresponding authors upon request.

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

## Acknowledgements

We acknowledge DESY (Hamburg, Germany), a member of the Helmholtz Association HGF, and Helmholtz-Zentrum Berlin for the provision of excellent experimental facilities. Parts of this research were carried out at PETRA III and BESSY II. Beamtime was allocated for proposals I-20210172 (PETRA III) and 202-09741 (BESSY II). We thank the P04 and U49-2_PGM−1 beamline staff for their excellent assistance. This work was supported by the German Federal Ministry of Education and Research (BMBF) through projects 05K19RK2—GPhase and 05K22RK1—TRANS-ALP. A.E., A.H., C.K.-W., D.B., E.H., and L.M. acknowledge support from SFB 1319 ELCH, funded by the Deutsche Forschungsgemeinschaft (DFG; project No. 328961117). We acknowledge the scientific exchange and support of the Centre for Molecular Water Science (CMWS). We thank André Knie for support at the early stage of this project. F.T. acknowledges funding by the Deutsche Forschungsgemeinschaft (DFG, German Research Foundation)—Project 509471550, Emmy Noether Programme and acknowledges support by the MaxWater initiative of the Max-Planck-Gesellschaft. L.S.C. gratefully acknowledges financial support by the European Research Council (ERC) (Advanced Investigator Grant No. 692657). O.B. acknowledges support from the Swedish Research Council through project 2023-04346.

## Author contributions

D.B. and A.H. conceived the experiments. D.B., F.T., I.U., C.Z., C.H., J.V., N.K., L.M., C.K.-W., E.H., U.H., and A.H. performed the experiments. D.B. carried out the analysis. D.B. and A.H. wrote the manuscript. D.B., F.T., I.U., L.S.C., O.B., U.H., A.E., and A.H. discussed the data and contributed to the writing of the manuscript.

## Funding

## Competing interests

The authors declare no competing interests.
