## [Peer Review File · Nature Communications]

X-ray radiation damage cycle of solvated inorganic ionsREVIEWER COMMENTS

Reviewer #1 (Remarks to the Author):

The manuscript titled "X-ray radiation damage cycle of solvated inorganic ions: low-energy electron emission" applies coincidence XPS experiments to investigate the electronic decay pathways following X-ray ionization of an ion in solution. The primary technical advance of this work is that it overcomes some of the challenges faced in photoelectron spectroscopy due to the typically highly-congested spectra, to isolate interatomic/intermolecular Coulombic decay (ICD) and electron-transfer-mediated decay (ETMD) processes.

The authors present a novel and detailed study on the radiation damage mechanisms at the molecular level, focusing on the production of low-energy electrons (LEEs) through a cascade of ETMD and ICD processes. By employing advanced multi-electron coincidence spectroscopy, the authors investigate the decay of 1s vacancies in solvated Mg^{2+} ions, providing novel insights into the radiation damage processes in aqueous environments. The study is well-structured, with clear objectives, rigorous experimental methods, detailed results, and a thoughtful discussion.

The experimental methods used are advanced and well established. Reproduction of data on Al^{3+} shows reliability and reproducibility of the method used. The work reported here is clearly motivated by, and investigates the decay mechanisms revealed in the recently published theoretical work in reference 7.

The paper is a valuable contribution to the field of radiation chemistry and the study of solvated inorganic ions under irradiation. The specific decay pathways, and timescales isolated here, are clearly of broad interest to the multidisciplinary readership of Nature Communications. With minor revisions, as itemized below, the paper would be suitable for publication. I recommend acceptance after minor revisions.

1. A highly-concentrated concentrated solution (3M of $MgCl_2$) was investigated in the experiments. Were other concentrations investigated for comparison? Please address any possible role of the chloride anion, including some discussion of any possible influence of chlorides near the Mg^{2+} ion and solvation shell, such as recombination processes and timescales, and its relevance with respect to the ETMD and the ICD channels.
2. Work described in ref. 7 considers water hexamer in the vicinity of Mg^{2+} : $Mg^{2+}(H_2O)_6$ and respective decay times. Please add some brief discussion about the restriction of this model solution to a single solvation shell, and the possible different decay times if more shells are included.
3. Please add an explanation about the gray (left) and black (right) arrows to the caption of Fig. 2

Reviewer #2 (Remarks to the Author):

Review report: D_Bloß_et_al

The manuscript titled "X-ray radiation damage cycle of solvated inorganic ions: low-energy electron emission" applies coincidence XPS experiments to investigate the electronic decay pathways following X-ray ionization of an ion in solution. The primary technical advance of this work is that it overcomes some of the challenges faced in photoelectron spectroscopy due to the typically highly-congested spectra, to isolate interatomic/intermolecular Coulombic decay (ICD) and electron-transfer-mediated decay (ETMD) processes.

The authors present a novel and detailed study on the radiation damage mechanisms at the molecular level, focusing on the production of low-energy electrons (LEEs) through a cascade of ETMD and ICD processes. By employing advanced multi-electron coincidence spectroscopy, the

authors investigate the decay of 1s vacancies in solvated Mg^{2+} ions, providing novel insights into the radiation damage processes in aqueous environments. The study is well-structured, with clear objectives, rigorous experimental methods, detailed results, and a thoughtful discussion.

The experimental methods used are advanced and well established. Reproduction of data on Al^{3+} shows reliability and reproducibility of the method used. The work reported here is clearly motivated by, and investigates the decay mechanisms revealed in the recently published theoretical work in reference 7.

The paper is a valuable contribution to the field of radiation chemistry and the study of solvated inorganic ions under irradiation. The specific decay pathways, and timescales isolated here, are clearly of broad interest to the multidisciplinary readership of Nature Communications. With minor revisions, as itemized below, the paper would be suitable for publication. I recommend acceptance after minor revisions.

A highly-concentrated concentrated solution (3M of $MgCl_2$) was investigated in the experiments. Were other concentrations investigated for comparison? Please address any possible role of the chloride anion, including some discussion of any possible influence of chlorides near the Mg^{2+} ion and solvation shell, such as recombination processes and timescales, and its relevance with respect to the ETMD and the ICD channels.

Work described in ref. 7 considers water hexamer in the vicinity of Mg^{2+} : $Mg^{2+}(H_2O)_6$ and respective decay times. Please add some brief discussion about the restriction of this model solution to a single solvation shell, and the possible different decay times if more shells are included.

Please add an explanation about the gray (left) and black (right) arrows to the caption of Fig. 2.

Reviewer #3 (Remarks to the Author):

The manuscript entitled "X-ray radiation damage cycle or solvated inorganic ions: low-energy electron emission" by D. Bloss et al. presents experimental results on the emission of low-energy electrons following core ionisation of Mg^{2+} ions solvated in water. Here the authors used an original approach coupling a magnetic bottle electron spectrometer thus allowing for coincidence measurements. Using the coincidence with the photoelectron emitted by core shell ionisation of the metal ion they are able to subtract the background in the low-energy range due to scattered electrons and therefore they can distinguish the features associated by decay mechanisms of the ionised Mg ions. These features are insensitive to photon energy supporting their assignation to ICD and ETMD decay mechanisms. The experimental method sound and the results seems strong. The authors compare their experimental results to a previous theoretical study by Cederbaum's group. While some experimental features can be assigned to specific mechanisms by comparison with the theoretical values, the agreement is not good enough for several features. The authors discuss this discrepancy by the fact that calculations are on clusters thus the Mg ions is not fully solvated contrary to the experiments using a liquid microjet. I have the feeling that the discussion may be developed further. I have also the feeling that the presentation of results is somehow strange. E.g. the presentation of ETMD following 2p ionisation is more detailed when presenting 2s ionisation (the highest estimated ETMD electron energy not discussed in the 2p section). In Fig4a one maximum is also visible around 10 eV but in text only the one at 19 eV is described. Is the first maximum due to the subtraction method? It is matching quite well with the feature observed after 2p ionisation.

I have also two minor comments:

- in Fig2, some arrows are grey but no reason are given in the text or caption
- in the fourth paragraph of discussion "the relaxation of a Mg^{3+} ($1s^{-1}$)"

Reviewer #4 (Remarks to the Author):

The authors measured multi-electron coincidence spectra after 2p, 2s, and 1s electron photoionization of Mg²⁺ from a 3 M aqueous MgCl₂ solution. They compared the results to the predicted decay cascade after Mg²⁺ 1s electron inner-shell ionization. I read with considerable interest this manuscript describing experiments, which I consider as state-of-the-art in the measurements of specific photoionization processes in liquids. In my opinion, such measurements are among the most, if not the most difficult to perform in the radiation sciences. These researchers found that the absorption of a single high-energy photon by a metal ion leads to an ultrafast, radiationless decay cascade producing several LEEs via ETMD and ICD that potentially may cause considerable local damage to the surrounding of the ion. This finding is of interest to explain the high efficiency of LEEs and local ionizations to induce DNA damage in radiation therapy and particularly in chemoradiation therapy. It stems mostly from the considerable details on the metal initial state they provide, showing that the cascade can start repeatedly and rapidly, largely multiplying the local damage, previously not estimated. On more speculative grounds, one can conjecture that the ionized water molecules could be transferred to further solvation shells following Coulomb explosion, i.e., to new water molecules diffusing into the surrounding of the metal ion, keeping the charge- and energy-transfer channels open and thus causing further local damage.

Based on my comprehension that the observed decay cascade is not exclusive to Mg ions and could proceed after ionization of other solvated metal ions, I hope that such measurements can be further pursued with water solutions of molecules and ions found in DNA. This could stand as a strong contribution to Radiobiology, considering that it is the cluster damages (i.e., multiple damages within 20 base pairs) that are by far the most detrimental to cell survival after high energy irradiation.

I do agree with the authors that interpretation of the experimental LEE spectra is challenging because of the high background of slow electrons. In trying to solve this problem, the authors applied highly efficient time-resolved techniques to measure coincidence spectra. Coincidences revealed significant structures in the LEE spectra, enabling assignments of individual spectral features to certain steps in the decay pathways. Such techniques are used by the authors because of the difficulty to obtain the information with other experimental techniques, due to multiple LEE inelastic scattering causing congested energy-loss spectra in the liquid phase. At this point, I would like to suggest that experiments with the same radiation source be conducted on thin films of amorphous ice deposited on a metal substrate. In this case, many individual LEE-energy losses can be resolved with a resolution around 0.005 eV with an appropriate high-resolution electron spectrometer, if the film is sufficiently thin, so as to reduce multiple electron energy losses. The data would not be recorded in the liquid phase, but I think it could further help in the interpretation of the present results. Liquid and amorphous states of water are in many respects similar and many of the differences are known.

Although X-ray-induced ICD and ETMD cascades are complex phenomena, I think the authors were able to explain them quite efficiently and link the data to the observed spectra for Mg²⁺. I get the impression that they have considered all possible processes of energy losses and all significant decay mechanisms. In summary, I find this work, performed by a group of experts in the field, to be highly credible and sound. It provides sufficient details for the reader to acquire a good understanding of the experiments and the results; the conclusions are a logical consequence of the results. It is difficult to further criticize these results and the experiments. My only comment relates to the time resolution of the experiments and the energy resolution of the electron spectrometer. These parameters are not clear to me; perhaps coincidence resolution and electron energy resolution are intertwined because both use time as the determinant factor. The situation would have been different if an electrostatic electron energy analyser had been used. Further explanations on resolution may be useful in this article. I suggest publication of the article once the authors have considered all the comments of this review.

Point-by-point reply to the reviewers' comments

Reviewers #1 and #2:

Comment 1.1: The manuscript titled “X-ray radiation damage cycle of solvated inorganic ions: low-energy electron emission” applies coincidence XPS experiments to investigate the electronic decay pathways following X-ray ionization of an ion in solution. The primary technical advance of this work is that it overcomes some of the challenges faced in photoelectron spectroscopy due to the typically highly-congested spectra, to isolate interatomic/intermolecular Coulombic decay (ICD) and electron-transfer-mediated decay (ETMD) processes.

The authors present a novel and detailed study on the radiation damage mechanisms at the molecular level, focusing on the production of low-energy electrons (LEEs) through a cascade of ETMD and ICD processes. By employing advanced multi-electron coincidence spectroscopy, the authors investigate the decay of 1s vacancies in solvated Mg²⁺ ions, providing novel insights into the radiation damage processes in aqueous environments. The study is well-structured, with clear objectives, rigorous experimental methods, detailed results, and a thoughtful discussion.

The experimental methods used are advanced and well established. Reproduction of data on Al³⁺ shows reliability and reproducibility of the method used. The work reported here is clearly motivated by, and investigates the decay mechanisms revealed in the recently published theoretical work in reference 7. The paper is a valuable contribution to the field of radiation chemistry and the study of solvated inorganic ions under irradiation. The specific decay pathways, and timescales isolated here, are clearly of broad interest to the multidisciplinary readership of Nature Communications. With minor revisions, as itemized below, the paper would be suitable for publication. I recommend acceptance after minor revisions.

Reply 1.1: We thank you very much for your time and effort and for this supportive report! We will address your points in detail below.

Comment 1.2: A highly-concentrated concentrated solution (3M of MgCl₂) was investigated in the experiments. Were other concentrations investigated for comparison? Please address any possible role of the chloride anion, including some discussion of any possible influence of chlorides near the Mg²⁺ ion and solvation shell, such as recombination processes and timescales, and its relevance with respect to the ETMD and the ICD channels.

Reply 1.2: We indeed studied different concentrations of the MgCl₂ solution. The data of Fig. S3 (a) in the Supplementary Information was measured with a concentration of 2 M and the data of Fig. S4 with a concentration of 4 M. All other spectra shown were recorded at a concentration of 3 M - including the spectra in the main article.

In principle, the Cl counter ion could participate in the decays. However, in earlier studies no contact ion pairing was found in the first solvation shell up to a concentration approaching saturation for the MgCl₂ (around 6 M). In our spectra, the observed features do not change with respect of the concentration for 2, 3, or 4 M. Therefore, we expect the contribution of the Cl anions to be negligible.

Changes 1.2: We added the following on page 5:

“In all steps of the cascade, besides ionization of neighboring water, the Cl counter ion could in principle participate in the decay. However, in earlier studies it was found that for Mg^{2+} in solution up to a concentration near the saturation for $MgCl_2$ no contact ion pairing can be found in the first solvation shell^{20,33,34}. Hence, we expect the contribution from Cl to the ICD and ETMD processes to be negligible.”

Comment 1.3: Work described in ref. 7 considers water hexamer in the vicinity of Mg^{2+} : $Mg^{2+}(H_2O)_6$ and respective decay times. Please add some brief discussion about the restriction of this model solution to a single solvation shell, and the possible different decay times if more shells are included.

Reply 1.3: Thank you for your comment. We tried to point out the restrictions of the model and additionally rearranged parts of our discussion.

Changes 1.3: We added and rearranged parts from a later position to page 4:

“While the calculations were performed on a $Mg^{2+}(H_2O)_6$ cluster and therefore considered only the first solvation shell, the full solvation in the experiment may lead to higher observable kinetic energies. This is due to larger polarization screening or charge separation over a wider range in the liquid phase beyond the first solvation shell²⁹. It has been observed earlier that in the liquid phase the screening of charge can be quite efficient²⁹. ~~Yet, the extent of these effects would be surprisingly high and cannot be fully clarified from the present data.~~ Another important aspect is that in the calculations of Ref. 7 the environment of the Mg has been assumed to be neutral for each step. In a real cascade, however, the already produced H_2O^+ cations may influence the further steps to a certain extent, until they will be replaced by neutrals from the environment. The presence of full solvation may also influence the decay widths of the excited states, although energy and charge exchange with the first solvation shell have been found to dominate for other systems³⁰.

Surprisingly, there is a good agreement between the estimated high-energy cutoff of ~ 33.2 eV neglecting the Coulomb repulsion for the ETMD(3) electron e_E [indicated by a green arrow and labeled E^ in Fig. 3(b)] and the high-energy cutoff in the experimental spectra around at least 30 eV in the double-electron coincidences. A similar observation was reported for the spectra of the decay of inner-shell-ionized Al^{3+} , see Ref. 5. This is a strong evidence that not only the first solvation shell participates in the decay and therefore the charge may be delocalized or screened effectively by the extended environment^{31,32}.”*

Comment 1.4: Please add an explanation about the gray (left) and black (right) arrows to the caption of Fig. 2

Reply 1.4: Thank you for your comment.

Changes 1.4: We added the explanation in the caption on page 11:

“The gray arrows indicate a minor decay pathway, while the black arrows correspond to the main decay pathways after 1s ionization.”

Reviewer #3:

Comment 3.1: The manuscript entitled "X-ray radiation damage cycle or solvated inorganic ions: low-energy electron emission" by D. Bloss et al. presents experimental results on the emission of low-energy electrons following core ionisation of Mg^{2+} ions solvated in water. Here the authors used an original approach coupling a magnetic bottle electron spectrometer thus allowing for coincidence measurements. Using the coincidence with the photoelectron emitted by core shell ionisation of the metal ion they are able to subtract the background in the low-energy range due to scattered electrons and therefore they can distinguish the features associated by decay mechanisms of the ionised Mg ions. These features are insensitive to photon energy supporting their assignation to ICD and ETMD decay mechanisms. The experimental method sound and the results seems strong.

Reply 3.1: We thank you very much for your time and effort and for this supportive report! We will address all of your points in detail below.

Comment 3.2: The authors compare their experimental results to a previous theoretical study by Cederbaum's group. While some experimental features can be assigned to specific mechanisms by comparison with the theoretical values, the agreement is not good enough for several features. The authors discuss this discrepancy by the fact that calculations are on clusters thus the Mg ions is not fully solvated contrary to the experiments using a liquid microjet. I have the feeling that the discussion may be developed further. I have also the feeling that the presentation of results is somehow strange. E.g. the presentation of ETMD following 2p ionisation is more detailed when presenting 2s ionisation (the highest estimated ETMD electron energy not discussed in the 2p section).

Reply 3.2: Thank you for your input. We agree that our presentation can be improved by rearranging our discussion in this part. We also added a few parts to the discussion considering the role of the Cl counter ion (see also comments 1.2 and 1.3).

Changes 3.2: We added parts to the discussion and rearranged parts from a later position to page 4. See Comment 1.3.

Comment 3.3: In Fig4a one maximum is also visible around 10 eV but in text only the one at 19 eV is described. Is the first maximum due to the subtraction method? It is matching quite well with the feature observed after 2p ionisation.

Reply 3.3: The maximum around 10 eV is indeed the analogue to the feature observed after 2p ionization, since the 2p ionized state can be reached after 2s ionization and ICD. Two aspects are important for its interpretation. (1) Its kinetic energy may be affected by the presence of a H_2O^+ cation produced through

the preceding ICD step. (2) Due to an applied retardation voltage all electrons with even lower kinetic energies are repelled and cannot be detected. It therefore cannot be deduced from the present data whether this feature has indeed a maximum at around 10 eV or in reality extends further towards lower kinetic energies.

Changes 3.3: We added on page 5:

“The feature around 10 eV is mainly the analogue to the feature at equal kinetic energies in Fig. 3(b). It is expected to appear due to ETMD(2) from 2p ionized states which are populated by ICD of the 2s ionized states. Its low-energy onset cannot reliably be deduced from the present data due to the applied retardation voltage.”

Comment 3.4: I have also two minor comments:

- in Fig2, some arrows are grey but no reason are given in the text or caption
- in the fourth paragraph of discussion "the relaxation of a Mg^{3+} ($1s^{-1}$)"

Reply 3.4: Thank you for your comments. We corrected both points.

Changes 3.4: We added the explanation in the caption on page 11 and corrected the typo on page 5:

“The gray arrows indicate a minor decay pathway, while the black arrows exhibit the main decay pathway after 1s ionization.”

“relaxation of a Mg^{3+} ($1s^{-1}$)”

Reviewer #4:

Comment 4.1: The authors measured multi-electron coincidence spectra after 2p, 2s, and 1s electron photoionization of Mg^{2+} from a 3 M aqueous $MgCl_2$ solution. They compared the results to the predicted decay cascade after Mg^{2+} 1s electron inner-shell ionization. I read with considerable interest this manuscript describing experiments, which I consider as state-of-the art in the measurements of specific photoionization processes in liquids. In my opinion, such measurements are among the most, if not the most difficult to perform in the radiation sciences. These researchers found that the absorption of a single high-energy photon by a metal ion leads to an ultrafast, radiationless decay cascade producing several LEEs via ETMD and ICD that potentially may cause considerable local damage to the surrounding of the ion. This finding is of interest to explain the high efficiency of LEEs and local ionizations to induce DNA damage in radiation therapy and particularly in chemoradiation therapy. It stems mostly from the considerable details on the metal initial state they provide, showing that the cascade can start repeatedly and rapidly, largely multiplying the local damage, previously not estimated. On more speculative grounds, one can conjecture that the ionized water molecules could be transferred to further solvation shells following Coulomb explosion, i.e., to new water molecules diffusing into the surrounding of the metal ion, keeping the charge- and energy-transfer channels open and thus causing further local damage. Based on my comprehension that the observed decay cascade is not exclusive to Mg ions and could proceed after ionization of other solvated metal ions, I hope that such measurements can be further

pursued with water solutions of molecules and ions found in DNA. This could stand as a strong contribution to Radiobiology, considering that it is the cluster damages (i.e., multiple damages within 20 base pairs) that are by far the most detrimental to cell survival after high energy irradiation.

I do agree with the authors that interpretation of the experimental LEE spectra is challenging because of the high background of slow electrons. In trying to solve this problem, the authors applied highly efficient time-resolved techniques to measure coincidence spectra. Coincidences revealed significant structures in the LEE spectra, enabling assignments of individual spectral features to certain steps in the decay pathways. Such techniques are used by the authors because of the difficulty to obtain the information with other experimental techniques, due to multiple LEE inelastic scattering causing congested energy-loss spectra in the liquid phase.

Reply 4.1: We thank you very much for your time and effort and for this supportive report! We will address your points in detail below.

Comment 4.2: At this point, I would like to suggest that experiments with the same radiation source be conducted on thin films of amorphous ice deposited on a metal substrate. In this case, many individual LEE-energy losses can be resolved with a resolution around 0.005 eV with an appropriate high-resolution electron spectrometer, if the film is sufficiently thin, so as to reduce multiple electron energy losses. The data would not be recorded in the liquid phase, but I think it could further help in the interpretation of the present results. Liquid and amorphous states of water are in many respects similar and many of the differences are known.

Reply 4.2: Thank you for your suggestions and further input. While this idea is certainly interesting, we also see major challenges. For example, the LEE background is typically also present from solid-state samples. A coincident electron detection from thin films in the way we conducted it, has, to our best knowledge, not been realized so far. We will consider if future experiments may be possible.

Comment 4.3: Although X-ray-induced ICD and ETMD cascades are complex phenomena, I think the authors were able to explain them quite efficiently and link the data to the observed spectra for Mg²⁺. I get the impression that they have considered all possible processes of energy losses and all significant decay mechanisms. In summary, I find this work, performed by a group of experts in the field, to be highly credible and sound. It provides sufficient details for the reader to acquire a good understanding of the experiments and the results; the conclusions are a logical consequence of the results. It is difficult to further criticize these results and the experiments. My only comment relates to the time resolution of the experiments and the energy resolution of the electron spectrometer. These parameters are not clear to me; perhaps coincidence resolution and electron energy resolution are intertwined because both use time as the determinant factor. The situation would have been different if an electrostatic electron energy analyser had been used. Further explanations on resolution may be useful in this article. I suggest publication of the article once the authors have considered all the comments of this review.

Reply 4.3: The energy resolution of a magnetic-bottle-type time-of-flight spectrometer depends mainly on the length of the drift tube. For similar spectrometers a resolution of around $E/\Delta E \sim 30$ was determined.

Note that this experiment is not time-resolved with respect to the physical processes of interest, which happen on a femtosecond timescale. This timescale is not directly accessible at synchrotron light sources due to the typical pulse length of about 100 ps. Nevertheless, the coincident emission of electrons can be deduced from their time-of-flight, which is in the nanosecond regime.

We agree that the overall energy resolution of electrostatic electron analyzers is in general significantly better than that of a time-of-flight spectrometer. It has, however, not the capability of detecting two electrons of considerably different kinetic energies in coincidence.

Changes 4.3: We added a paragraph on the energy resolution in the Methods section on page 6.

“Similar magnetic bottle time-of-flight spectrometers have a resolution of $E/\Delta E$ around 30⁴⁰. The temporal resolution of the experiment is determined by the duration of the synchrotron pulses, which is around 100 ps.”